# Land Use and Land Cover Change in the Vaal Dam Catchment, South Africa: A Study Based on Remote Sensing and Time Series Analysis

Altayeb Obaid [1,*], Elhadi Adam [1] and K. Adem Ali [1,2]

1   School of Geography, Archeology and Environmental Studies, University of the Witwatersrand, Private Bag X3, P.O. Box Wits, Johannesburg 2050, South Africa
2   Department of Geology and Environmental Geosciences, College of Charleston, Charleston, SC 29466, USA
*   Correspondence: 2126105@students.wits.ac.za

**Abstract:** Understanding long-term land use/land cover (LULC) change patterns is vital to implementing policies for effective environmental management practices and sustainable land use. This study assessed patterns of change in LULC in the Vaal Dam Catchment area, one of the most critically important areas in South Africa, since it contributes a vast portion of water to the Vaal Dam Reservoir. The reservoir has been used to supply water to about 13 million inhabitants in Gauteng province and its surrounding areas. Multi-temporal Landsat imagery series were used to map LULC changes between 1986 and 2021. The LULC classification was performed by applying the random forest (RF) algorithm to the Landsat data. The change-detection analysis showed grassland being the dominant land cover type (ranging from 52% to 57% of the study area) during the entire period. The second most dominant land cover type was agricultural land, which included cleared fields, while cultivated land covered around 41% of the study area. Other land use types covering small portions of the study area included settlements, mining activities, water bodies and woody vegetation. Time series analysis showed patterns of increasing and decreasing changes for all land cover types, except in the settlement class, which showed continuous increase owing to population growth. From the study results, the settlement class increased considerably for 1986–1993, 1993–2000, 2000–2007, 2007–2014 and 2014–2021 by 712.64 ha (0.02%), 10245.94 ha (0.26%), 3736.62 ha (0.1%), 1872.09 ha (0.05%) and 3801.06 ha (0.1%), respectively. This study highlights the importance of using remote sensing techniques in detecting LULC changes in this vitally important catchment.

**Keywords:** remote sensing; image classification; random forest; change detection

## 1. Introduction

Land resources have been used for social, material and cultural human demands, which leads to significant changes in LULC patterns [1,2]. Such changes have consequently been associated with various impacts and effects on different fields at multiple scales; local, regional and global [3,4], including surface energy balance through its effect on the weather and climate at local, regional [5,6] and global scales, such as how changing the tropical rainforest areas impacts the global climate [7]. LULC changes affect the hydrological cycle by altering the hydrological response of watersheds in terms of surface runoff, decreasing groundwater recharge, water quality and pollutant transfers. These factors affect the dispersion of non-point water pollutants and direct them into freshwater bodies [8]. Furthermore, changes in LULC affect biodiversity and aquatic systems [2,4,9,10] and the receiving ecosystem service values by affecting their structure and functioning [11,12]. Therefore, understanding the spatial and temporal variations of LULC on a watershed level is critical for effective monitoring, planning and management of the resources and ecosystems [10]. Accurate information on LULC changes can also contribute to other applications, including the assessment of damage and deforestation, the monitoring disasters and measurement of

the expansion of urban areas. It also assists with land use management and planning [13]. However, obtaining such information can be achieved through performing long-term time series analysis of the LULC changes [1,13].

The Vaal Dam Catchment forms a vast part of the Upper Vaal Water Management Area (WMA). It is part of the Vaal drainage system in South Africa [14]. Extensive gold and coal mining activities have taken place within the Upper Vaal catchment [15]. In the Vaal Dam Catchment area, the range of land use includes major agricultural activities (encompassing mainly cattle grazing and dry land cultivation), gold and coal mining and some industrial activity [8,9]. In the past, most human activities within the Vaal Dam Catchment area were dependent on or related to agriculture [16–18]. The most extensively cultivated areas within the Vaal Dam Catchment have been near the Vaal and the lower-lying valleys of the Wilge River, with stock farming in the hilly parts [16]. With the building for Sasol of the synthetic fuel complex and the commencement of coal mining in late 1970 (alongside other economic developments), the land use character of the catchment has changed substantially. The towns related to these developments grew significantly, mainly in the Waterval sub-catchment in the northern part of the Vaal Dam Catchment [18]. These economic developments have contributed to the ongoing expansion of settlement areas in the catchment area [10]. The southern and southeastern part of the Vaal Dam Catchment is contained in the Wilge River Sub-catchment, which is dominated by agricultural land (consisting of non-cultivated arable land, cultivated farms, livestock pastures and some human settlement areas) [12]. As human activity has intensified in recent decades, ecosystems within the catchment have been degraded [18]. As a result of these activities, the discharge of treated effluents from the mine dewatering and urban areas within some areas of the Vaal Dam Catchment returns into the river system and causes significant impacts on river water quality [15].

A remote sensing (RS) approach is particularly effective for characterising the LULC changes for large areas such as the Vaal Dam Catchment. Satellite RS has been used as a cost-effective method in mapping and developing a clear understanding of LULC changes. Various satellites with moderate to high spatial resolution and temporal coverage are available that can be used to study long-term changes in LULC [19]. Many studies have used satellites such as the Moderate Resolution Imaging Spectroradiometer (MODIS) [20–22] and Landsat series data to study LULC at regional scales in different regions in the world [23]. MODIS coverage started in 2000 with a spatial resolution of one kilometre and a revisit time of one day. Starting in 1972, Landsat has had a longer coverage time; since then, a new era in earth observations has begun, using moderate (60-metre) spatial resolution imagery [24,25]. After 1982, Landsat sensors started to acquire data in higher (30-metre) spatial resolution but with a lower (16-day) temporal resolution. However, since LULC change is not noticeable over short periods (such as hours and days), Landsat is better at detecting LULC change owing to its higher spatial resolution and its comprehensive archive compared to the coarse resolution satellite data such as NASA's MODIS. The Landsat archive is very valuable for estimating area changes over time; it allows the LULC changes to be thoroughly assessed and statistically quantified [19]. Another advantage of using Landsat data is the consistency of configurations of the various generation sensors (TM, ETM+ and OLI) within visible to shortwave infrared bands, as well as their spatial resolution (30 m), which allows us to use a continuous data set starting in 1984 [24]. Obtaining information on LULC change based on RS has been used in many parts of the world to address various environmental challenges [12,19,24]. RS data and field observations can be combined to accomplish LULC classification and change detection. RS data provides faster processing and is more cost-effective than traditional methods [26]. Many studies conducted within the Vaal Dam Catchment area investigating water quality issues, few of them have highlighted the impact of LULC owing to human activities on surrounding water quality [18,27]. However, the studies conducted to date on LULC have lacked spatial and temporal resolution for accurately characterising the impact of LULC on the large-scale functioning of the ecosystem. The application of geospatial techniques can provide robust

methods for studying the impacts of LULC on large catchment systems such as the Vaal Dam Catchment [28]. The Vaal Dam Catchment is a primary water source for the Vaal Dam Reservoir beside the Lesotho highland water project, from which Rand Water supplies potable water to Gauteng province [29]. Many issues of water quality and ecosystem deterioration have been discussed in the literature for this crucial area [30]; understanding the LULC patterns and evaluating their effects on ecosystems and water quality are essential. To understand their dynamics, there is a need to first carry out LULC classification and then to detect respective changes in LULC. This will provide critical information that can be used for effective and sustainable environmental restoration and management of resources to avoid and minimise further deterioration in water quality and eco/aquatic systems. This research aims to accurately characterise the LULC change in the Vaal Dam Catchment area over recent decades (1986 to 2021) using NASA's Landsat data. It is anticipated that this could contribute towards evaluating requirements for better management of catchment.

## 2. Materials and Methods

### 2.1. Study Area

The Vaal Dam Catchment is located between 26.27° and 28.77° S, and 28.00° and 30.31° E, in the central plateau of South Africa. The annual rainfall generally ranges between 600 and 800 mm/y and occurs mostly between October and March during the summer season. In the south and southeastern part of the catchment, annual rainfall can reach up to 1500 mm/y [16,31]. The catchment has an area of about 38,000 km², with an altitude ranging between 1300 and 1850 m.a.s.l. The Vaal Dam Catchment consists of two main sub-catchments drained by two major river systems, the Vaal River and Wilge River catchments (Figure 1). The two river systems are the main arteries for the Vaal Dam Catchment that supplies fresh water to the Vaal Dam Reservoir, from which water is supplied to Gauteng province and its surrounding areas. The southern and southeastern parts of the catchment are mountainous areas, and they are the highest parts of the catchments. In this part, the Wilge River rises from the northern slopes of Mont-aux-Sources in the Drakensberg mountain range, whereas the northern and northeastern parts of the catchment are relatively flat areas.

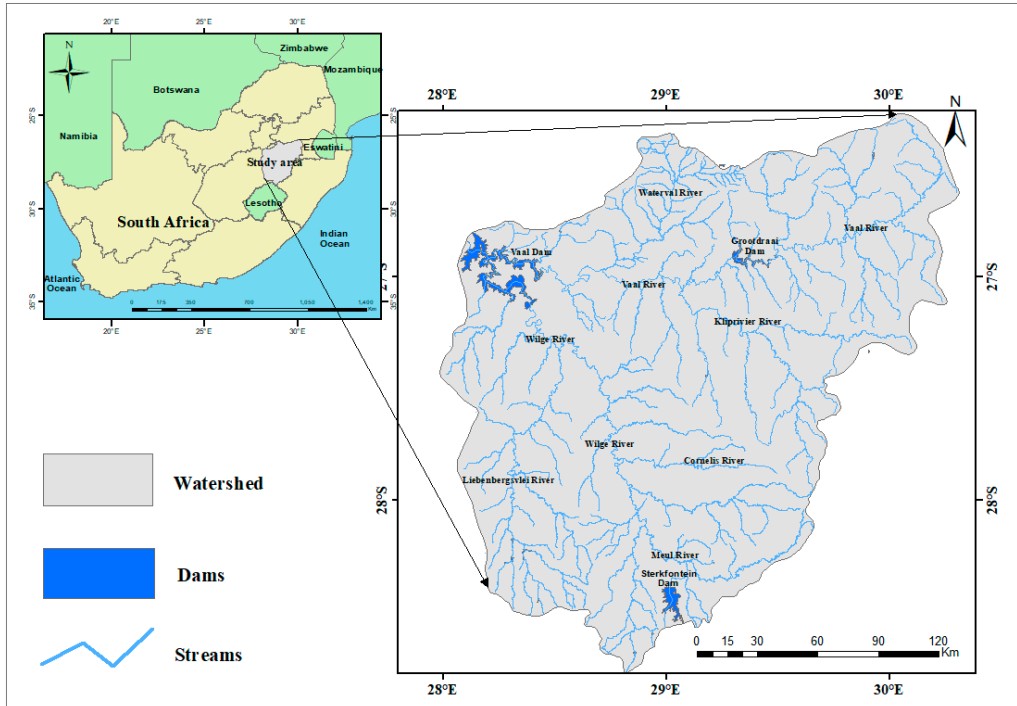

**Figure 1.** The location map of the study area.

The study area was chosen since the Vaal Dam is one of the most important water sources in South Africa and since there are considerable concerns regarding water quality in the catchment over recent decades. Further, there is interest from the water authorities in understanding the pattern of LULC changes in the catchment since this information will assist them in better understanding potential threats to the water quality of the dam.

### 2.2. Identification of LULC Types

Historical Google Earth images and available South African National Land Cover (SANLC) classification schemes were used as guides for the identification of the major LULC types in the study area (SANLC data sets for 1990, 2013–2014 and the latest data (SANLC-2018). The data sets can be obtained from the Department of Forestry, Fisheries and the Environment website (https://egis.environment.gov.za/sa_national_land_cover_datasets) accessed on 15 October 2020.

The SANLC-2018 data set uses 20-metre resolution Sentinel-2 imagery. The study made use of seven major LULC classes, including agriculture, cleared fields, grassland, mining, settlements, water bodies and woody vegetation (Table 1).

**Table 1.** Land cover classes used in this study.

| Class | Description |
|---|---|
| Agriculture | Crop areas that were green at the time of imagery |
| Cleared fields | Cleared fields, bare lands and shaded agricultural fields (greenhouses) |
| Grassland | Sparsely wooded and natural grassland areas plus fellow land and old fields |
| Mining | Ash piles, dumps and slimes dams accumulated from coal and gold mining |
| Settlements | Residential, industrial, commercial and mixed urban build-up areas |
| Water bodies | Rivers, open water, lakes, ponds and reservoirs |
| Woody vegetation | Natural, planted forests and orchards |

### 2.3. Data and Methods

2.3.1. Data Acquisition and Pre-Processing

Level 1 imageries were acquired from the USGS earth explorer platform (http://earthexplorer.usgs.gov/) accessed on 13 June 2021. A total of 36 cloud-free or good-quality images of Landsat 5 (TM), Landsat 7 (ETM+) and Landsat 8 (OLI) were obtained for the years 1986, 1993, 2000, 2007, 2014 and 2021 to produce 7-year-interval LULC maps. For each year, six scenes were required to cover the span of the Vaal Dam Catchment, and their respective paths/rows were specified (Table 2).

**Table 2.** The Landsat satellite images used in the study.

| Year | Satellite | Sensor | Path/Row | Resolution (m) | Acquisition Date |
|---|---|---|---|---|---|
| 1986 | Landsat 5 | TM | 169/078, 169/079, 169/080 | 30 | 20.05.1986 |
| | Landsat 5 | TM | 170/078, 170/079, 170/080 | 30 | 11.05.1986 |
| 1993 | Landsat 5 | TM | 169/078, 169/079, 169/080 | 30 | 24.06.1993 |
| | Landsat 5 | TM | 170/078,170/079, 170/080 | 30 | 30.05.1993 |
| 2000 | Landsat 7 | ETM+ | 169/078, 169/079, 169/080 | 30 | 22.08.2000 |
| | Landsat 7 | ETM+ | 170/078, 170/079, 170/080 | 30 | 28.07.2000 |
| 2007 | Landsat 5 | ETM+ | 169/078, 169/079, 169/080 | 30 | 06.05.2007 |
| | Landsat 5 | ETM+ | 170/078, 170/079, 170/080 | 30 | 29.05.2007 |
| 2014 | Landsat 8 | OLI | 169/078, 169/079, 169/080 | 30 | 17.05.2014 |
| | Landsat 8 | OLI | 170/078, 170/079, 170/080 | 30 | 24.05.2014 |
| 2021 | Landsat 8 | OLI | 169/078, 169/079, 169/080 | 30 | 04. 05.2021 |
| | Landsat 8 | OLI | 170/078, 170/079, 170/079 | 30 | 11.05.2021 |

Radiometric calibration was applied to each scene, followed by atmospheric correction using Fast Line-of-sight Atmospheric Analysis of Spectral Hypercube (FLAASH) module in ENVI (v5.4). The scenes of each year were mosaicked together using a seamless mosaic in ENVI to produce a single image for each year before cropping them to the study area boundaries using a shape file that had been prepared for this purpose by digitising the watershed boundary following the water divide based on stream network of the watershed.

The gap-fill module attached to ArcGIS was used to fill the gaps from Landsat 7 ETM+ scan line corrector failure. Images were then ready for LULC classification.

### 2.3.2. Reference Data

The reference dataset was obtained through visual interpretation of the pre-processed Landsat data sets. Well-distributed training areas were selected to ensure adequate representation of each class within the entire image extension. Point features with class labels for all classes were digitised on all the mosaicked images using ArcGIS. The point features of each year's image were then loaded on historical high-resolution Google Earth Pro imagery (http://earth.google.com/) accessed on 23 November 2021 and the temporal data was used to assess consistency. Points that fell in the area of inconsistent LULC were excluded from the analysis. Then, pixel values were extracted using the feature extraction tool in ArcGIS. An approximately equal number (100) of sample points was obtained for all classes, and, for the poorly separated classes, more representative point samples were added. The extracted pixel values of each class were randomly divided into training and validation data sets, 70% as the training data set used to train the classifier and 30% as the validation data set to test the classification accuracy. Table 3 shows the number of extracted reference data sets of each year used in the classification.

**Table 3.** The reference data numbers used in this study. Training data (Tr), test data (Te) and total number of data points (To).

| LULC Class | 1986 | | | 1993 | | | 2000 | | | 2007 | | | 2014 | | | 2021 | | |
|---|---|---|---|---|---|---|---|---|---|---|---|---|---|---|---|---|---|---|
| | Tr | Te | To | Tr | Te | To | Tr | Te | To | Tr | Te | To | Tr | Te | To | Tr | Te | To |
| Agriculture | 67 | 28 | 95 | 70 | 30 | 100 | 70 | 30 | 100 | 63 | 27 | 90 | 70 | 30 | 100 | 70 | 30 | 100 |
| Cleared field | 182 | 78 | 260 | 108 | 46 | 154 | 192 | 81 | 273 | 132 | 56 | 188 | 160 | 68 | 228 | 140 | 60 | 200 |
| Grassland | 70 | 30 | 100 | 70 | 30 | 100 | 72 | 30 | 102 | 70 | 30 | 100 | 70 | 30 | 100 | 70 | 30 | 100 |
| Mining | 70 | 30 | 100 | 70 | 30 | 100 | 56 | 24 | 80 | 50 | 21 | 71 | 70 | 30 | 100 | 70 | 30 | 100 |
| Settlements | 70 | 30 | 100 | 70 | 30 | 100 | 70 | 29 | 99 | 70 | 30 | 100 | 70 | 30 | 100 | 70 | 30 | 100 |
| Water bodies | 87 | 37 | 124 | 70 | 30 | 100 | 70 | 30 | 100 | 70 | 30 | 100 | 70 | 30 | 100 | 70 | 30 | 100 |
| Woody vegetation | 70 | 30 | 100 | 70 | 30 | 100 | 67 | 28 | 95 | 70 | 30 | 100 | 70 | 30 | 100 | 70 | 30 | 100 |

### 2.3.3. Image Classification

In this study, the random forest (RF) model was used as the LULC classifier. The model is an ensemble-based classification algorithm suggested by Breiman in 2001 to improve the performance of classification and regression trees (CART) [32]. It combines a large set of decision trees. The algorithm uses bagging and random selection techniques to build several binary classification trees (*ntree*) by using bootstrap samples with replacements driven from the original training data sets (each bootstrap sample produces a tree, and *ntree* is grown from bootstrap samples) [33]. Then, each predicted classification tree contributes a single vote to assign the most frequent class for the input data. The classifier output is determined by the majority of tree votes. If a separate test data set is not available, the out-of-bag (OOB) method can be used; around one-third of the samples are left out randomly for each newly generated training data set, called OOB samples. The OOB samples are used to measure the variable importance and estimate the misclassification errors [34,35]. In RF classification algorithm, two parameters need to be defined, *mtry*

(the number of variables to split at each node) and *ntree* (the number of trees to grow). A given number of input variables (*mtry*) at each node are randomly chosen from a random feature subset and the best split is then calculated using only this subset of input features. However, to ensure lower similarity between individual trees and thus a low bias, the tree is allowed to grow fully without pruning [36]. RF parameters (*mtry* and *ntree*) have to be optimised to improve the classification accuracy [35,37]. Using the algorithm, the default number of decision trees (*ntree*) is set to 500, while the default value of the variables number (*mtry*) corresponds to the square root of the total number of predictor variables (spectral bands) used in the study [38,39]. Based on several studies, the RF algorithm is regarded as a robust machine-learning LULC classifier with higher performance [34,36,40–42]. RF has many advantages over other machine learning classifiers, in that it is less sensitive to outliers, noise and overtraining; it can also handle large data sets and gives estimations of the important variables in the classification, along with internal generalisation error (OOB error) estimates, in an unbiased fashion [34,40,43].

In this study, the classifications were separately performed using stacked images containing bands from 1 to 6 of TM/ETM+ images, and bands from 1 to 7 of OLI images. The *mtry* and *ntree* were kept as default values, the *mtry* was set to the number of variables which were the number of bands of each stacked image (6 for TM and ETM+ images and 7 for OLI images) and *ntree* was set to 500, and a repeated 10-fold cross-validation was used to obtain the parameters of RF optimisation using the training data set only. The RF algorithm was run using the caret package in R statistical software, and the best combination of *mtry* and *ntree* was obtained based on the lowest OOB error (Table 4).

**Table 4.** The combination of *mtry* and *ntree* used to train the model for image classification.

| Year | *mtry* | *ntree* | Best Performance |
|---|---|---|---|
| 1986 | 2 | 2500 | 0.08 |
| 1993 | 5 | 500 | 0.07 |
| 2000 | 5 | 500 | 0.08 |
| 2007 | 4 | 5500 | 0.06 |
| 2014 | 2 | 500 | 0.09 |
| 2021 | 4 | 1500 | 0.07 |

2.3.4. Accuracy Assessment

To evaluate the quality of the thematic LULC maps developed from Landsat images using RF classifier, accuracy measures were calculated for each year using an independent set of data obtained from the reference data.

A confusion matrix was generated for each classified image using the test data set to quantify the classification accuracy of the RF performance based on the kappa index, overall accuracy (OA), producer accuracy (PA) and user accuracy (UA). The kappa index is a measure of how the classification results compare to values assigned by chance. Its values are between 0 and 1. If the kappa coefficient equals 0, it means no agreement between the classified image and the reference image. If the kappa coefficient equals 1, then the classified image and the reference image are identical. Thus, the higher the kappa coefficient is, the more accurate the classification. OA represents the percentages between the total of pixels correctly classified for all classes and the total number of pixels used in the data set. While PA is the number of correctly identified pixels divided by the total number of pixels in the reference image, UA is the number of the correctly identified pixels of a class, divided by the total number of pixels of the class in the classified image.

The flowchart below (Figure 2) outlines the steps of the classification method and change detection.

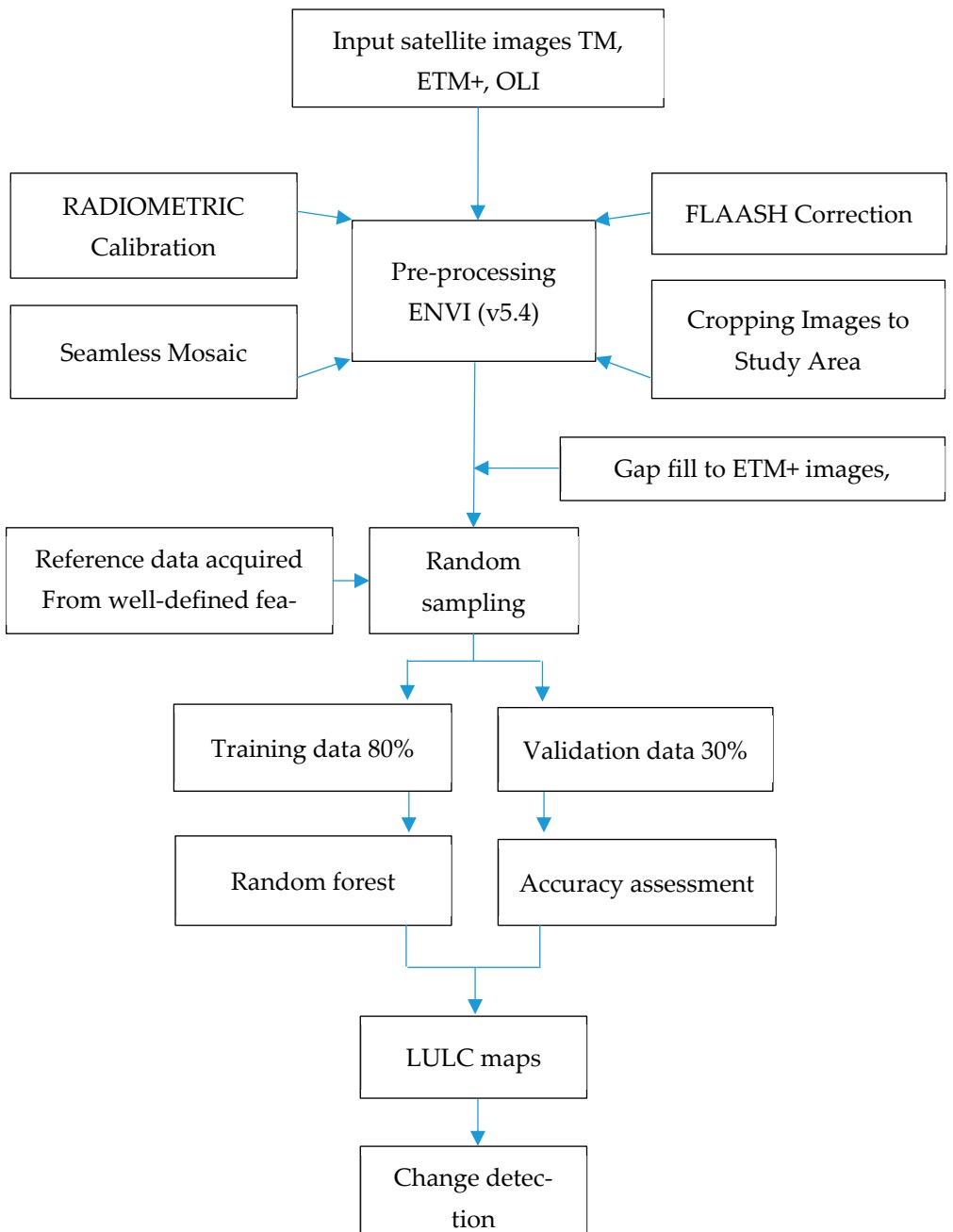

**Figure 2.** The methodology flow chart for the study to outline the classification and change detection steps.

### 3. Results

*3.1. Spatial Pattern of LULC Classes (1986 to 2021)*

Seven LULC classes were extracted from the processed images of the study area. Referring to Figure 3 below, the spatial distribution of the LULC classes from 1986 to 2021 was as follows: the grassland land cover type was generally dominant in the study area throughout the study period. This was followed by the lands used for crop cultivation activities, with land uses including cleared fields and agricultural land cover types. The grasslands covered between 52.0 per cent in 2007 to 56.9 per cent of the total study area in 2021, followed by cleared fields ranging between 28.6 in 2021 and 41.8 per cent in 1986. Agriculture was 1.41 per cent in 1986 and ranged to 10.0 per cent in 2021. The remaining classes covered small portions of the total area. Mining ranged between 0.14 in 1986 and 0.39 in 2021, settlements varied between 0.87 in 1986 and 1.43 in 2021, while water bodies

ranged from 0.65 in 1986 to 2.10 in 2007, and woody vegetation was at 2.66 in 1986 and decreased to 0.29 in 2000.

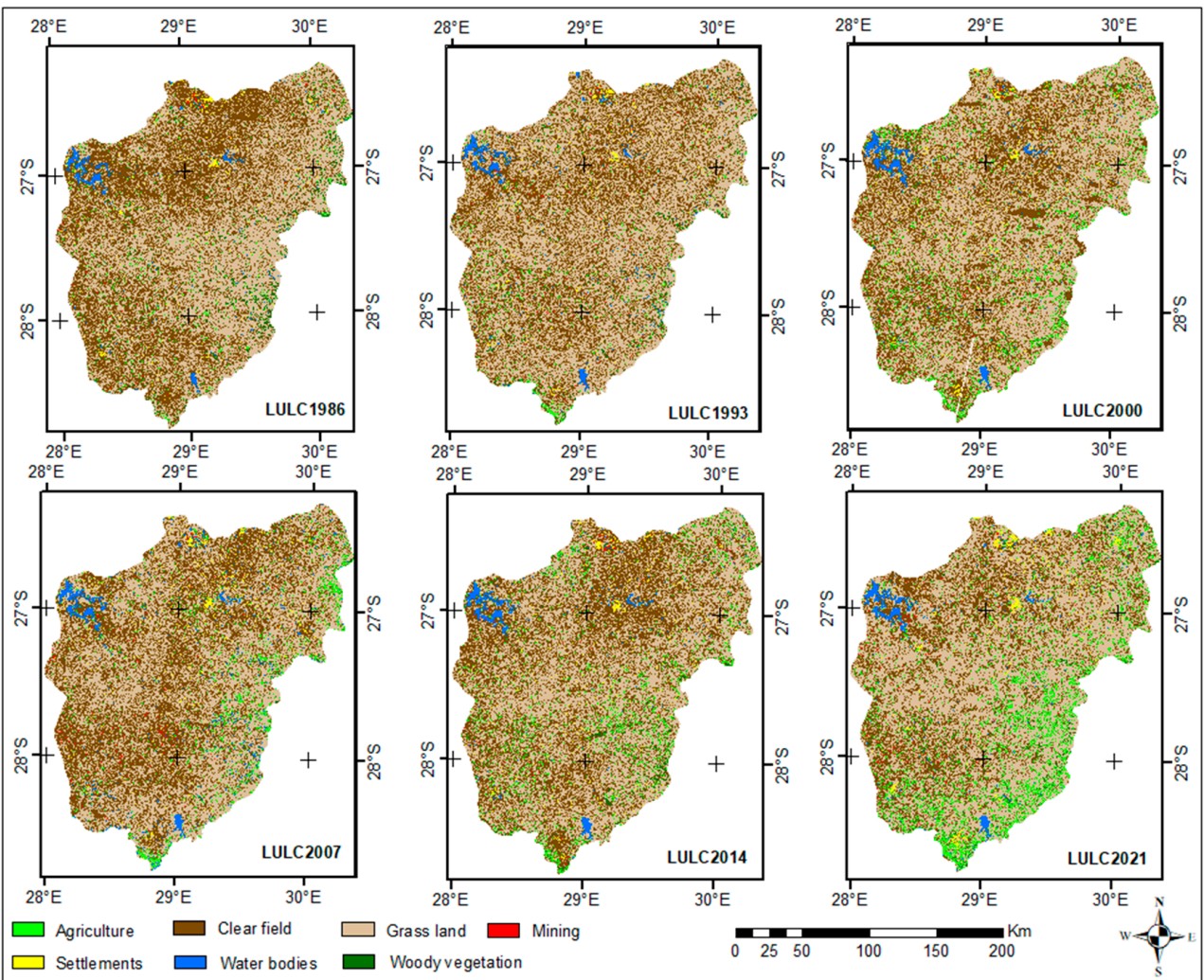

**Figure 3.** LULC patterns in the Vaal Dam Catchment from 1986 to 2021 in 7-year intervals.

The LULC classifications based on six different images were obtained based on the best combination of *mtry* and *ntree* parameters in RF. Table 4 above shows the selected combinations of the two input parameters used to train the classifier for each year according to their best performance. Table 5 summarises the class areas in hectares LULC patterns for 1986, 1993, 2000, 2007, 2014 and 2021.

**Table 5.** The LULC class areas in hectares and their relative proportions, based on the total area.

| Year | 1986 | | 1993 | | 2000 | | 2007 | | 2014 | | 2021 | |
|---|---|---|---|---|---|---|---|---|---|---|---|---|
| **LULC Area (Hectare)** | **Area** | **%** | **Area** | **%** | **Area** | **%** | **Area** | **%** | **Area** | **%** | **Area** | **%** |
| Agriculture | 54,605.88 | 1.41 | 111,290.7 | 2.88 | 176,796.5 | 4.58 | 133,960.9 | 3.46 | 166,057.5 | 4.29 | 387,083.0 | 9.98 |
| Cleared fields | 1,618,476 | 41.75 | 1,442,757 | 37.28 | 1,411,707 | 36.5 | 1,566,830 | 40.48 | 1,480,856 | 38.3 | 1,107,549.3 | 28.6 |
| Grasslands | 2,035,750 | 52.52 | 2,186,781 | 56.51 | 2,151,995 | 55.7 | 2,014,450 | 52.04 | 2,058,324 | 53.2 | 2,206,309.3 | 56.9 |
| Mining | 5545.08 | 0.14 | 10,179 | 0.26 | 8687.07 | 0.22 | 12,219.12 | 0.32 | 7,996.32 | 0.21 | 14,996.07 | 0.39 |
| Settlements | 33,539.67 | 0.87 | 34,252.31 | 0.89 | 44,498.25 | 1.15 | 48,234.87 | 1.25 | 50,107.77 | 1.29 | 55,361.52 | 1.43 |
| Water bodies | 25,198.56 | 0.65 | 34,653.87 | 0.9 | 58,830.57 | 1.52 | 81,455.49 | 2.10 | 50,914.08 | 1.32 | 55,778.85 | 1.44 |
| Woody vegetation | 103,116.5 | 2.66 | 49,653.36 | 1.28 | 11,214 | 0.29 | 13,881.78 | 0.36 | 56,395.35 | 1.46 | 52,673.22 | 1.36 |
| Total | 3,876,232 | 100 | 3,869,568 | 100 | 3,863,728 | 100 | 3,871,032 | 100 | 3,870,651 | 100 | 3,879,751 | 100 |

### 3.2. The Accuracy Assessment

The accuracy assessment for RF classifier was performed to evaluate the predicted performance of the trained model using the test data set; this was determined to be 30% of each data set. A confusion matrix for each classified image was then driven. The results of LULC classification indicated that the overall LULC classification accuracies (OA) for the six different date-classified images ranged from 87% in the 2014 classified image to 95% in the 2007 image, with kappa agreement indices ranging between 0.79 in 2014 and 0.92 for 2007. The user accuracy (UA) and producer accuracy (PA) are shown in Table 6. The performance of the internal classifier was good, with low OOB errors ranging between 3.75% and 8.98%. All classes had high user and producer accuracies ranging between 80 and 100, except for the settlement classes in 1986 and 2014; the UAs were 79% and 66%, respectively, and, in 2014 and 2021, the PAs were 59% and 63%, respectively. However, the nature of some classes and the different date-stacked images from neighbouring paths made it challenging to separate features in some classes owing to their similarity in their spectral signatures. For example, there was confusion between settlements and the shaded fields (greenhouses), with both containing spectrally similar features. This is also the case between settlement areas surrounded by trees and the woody vegetation class. The kappa values of the six classification results are sufficient for the study area because they satisfy the minimum 85% accuracy.

**Table 6.** The producer accuracies (PAs) and user accuracies (UAs) of the LULC classifications for the Vaal Dam.

|  | 1986 | | 1993 | | 2000 | | 2007 | | 2014 | | 2021 | |
| --- | --- | --- | --- | --- | --- | --- | --- | --- | --- | --- | --- | --- |
| LULC | PA% | UA% | PA% | UA% | PA% | UA% | PA% | UA% | PA% | UA% | PA% | UA% |
| Agriculture | 89 | 93 | 87 | 93 | 90 | 93 | 93 | 88 | 93 | 100 | 93 | 90 |
| Cleared fields | 86 | 91 | 91 | 91 | 96 | 92 | 80 | 89 | 85 | 80 | 94 | 84 |
| Grassland | 93 | 93 | 93 | 88 | 93 | 93 | 100 | 100 | 97 | 88 | 91 | 91 |
| Mining | 93 | 80 | 90 | 100 | 96 | 92 | 97 | 97 | 80 | 89 | 88 | 88 |
| Settlement | 87 | 79 | 97 | 97 | 86 | 96 | 97 | 100 | 59 | 66 | 63 | 86 |
| Water | 96 | 100 | 100 | 100 | 100 | 100 | 100 | 100 | 100 | 100 | 100 | 100 |
| Woody vegetation | 93 | 97 | 100 | 91 | 93 | 96 | 100 | 100 | 100 | 97 | 93 | 97 |

The RF classifier generally performs well in obtaining the seven determined classes in the Vaal Dam Catchment, as shown in the OA above. These results are the bases for the subsequent analysis of LULC change detections.

### 3.3. LULC Changes (1986 to 2021)

The ratio of each LULC class to the total area within the catchment varies per LULC class, with both increasing and decreasing trends being seen. Table 7 shows the pattern of LULC changes during the period studied (1986 to 2021).

The agriculture class showed an increasing pattern, except for the area change between 2000 and 2007. At the same time, cleared fields showed a decreasing pattern except between 2000 and 2007. The sum of the agriculture and cleared field areas indicates that the areas used in crop cultivation activities covered around 41 per cent +/−3% of the total study area.

The grassland areas show different patterns of increasing and decreasing in the size of their relative area. In general, they did not show a significant change, and grassland consistently remained the dominant class with little variation in extent in the study area.

**Table 7.** Area and percentage changes of the land use land cover (LULC) in the Vaal Dam Catchment for 1986, 1993, 2000, 2007, 2014 and 2021 images.

| Year | 1986–1993 | | 1993–2000 | | 2000–2007 | | 2007–2014 | | 2014–2021 | |
|---|---|---|---|---|---|---|---|---|---|---|
| LULC Change (In Hectares) | Area | % | Area | % | Area | % | Area | % | Area | % |
| Agriculture | 56,684.82 | +1.46 | 65,505.8 | +1.69 | −42,835.6 | −1.11 | 32,096.6 | +0.83 | 221,025.5 | +5.71 |
| Cleared Fields | −175,719 | −4.53 | −31,050 | −0.80 | 155,123 | +4.01 | −85,974 | −2.22 | −423307 | −10.94 |
| Grassland | 151,031 | +3.90 | −34,786 | −0.90 | −137,545 | −3.56 | +43,874 | +1.13 | 197,985.6 | +5.11 |
| Mining | 4633.92 | +0.12 | −1491.93 | −0.04 | 3532.05 | +0.09 | −4222.8 | −0.11 | 6999.75 | +0.18 |
| Settlements | 712.64 | +0.02 | 10,245.94 | +0.26 | 3736.62 | +0.10 | +1872.9 | +0.05 | 5253.75 | +0.14 |
| Water Bodies | 9455.31 | +0.24 | 24,176.7 | +0.62 | 22624.92 | +0.59 | −30,541.41 | −0.79 | 4864.77 | +0.13 |
| Woody Vegetation | −53,463.14 | −1.38 | −38,439.36 | −0.99 | 2667.78 | +0.07 | +42,513.57 | +1.10 | −3722.13 | −0.10 |

The settlement area showed expansion in the Vaal Dam Catchment from 1986 to 2021 Figure 4.

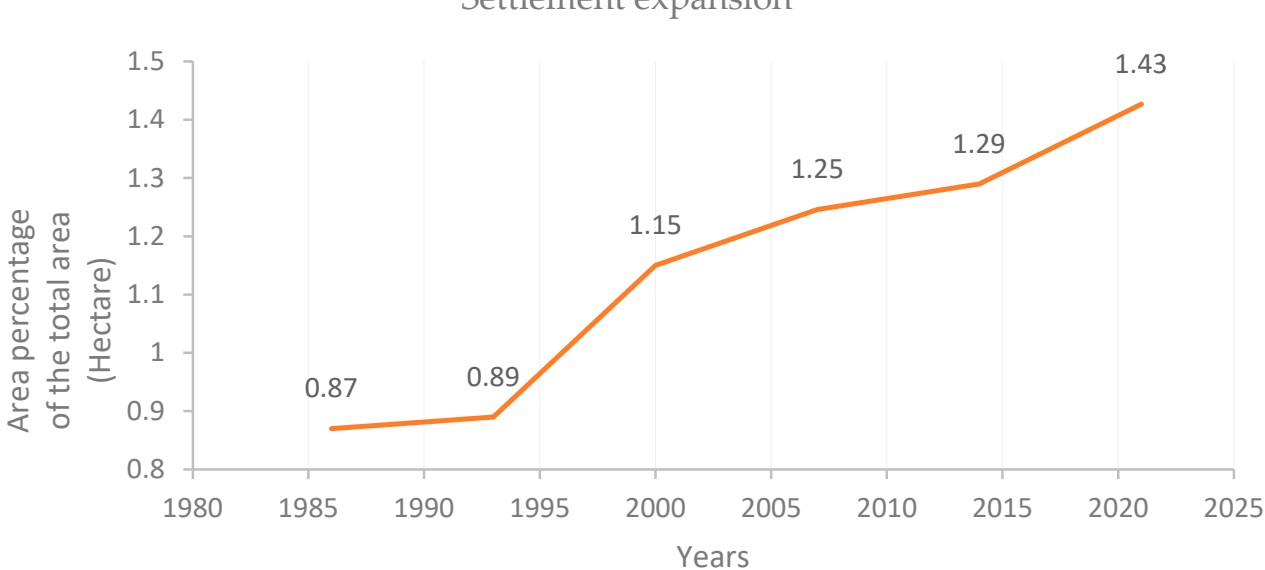

**Figure 4.** Settlement area changes between 1986 and 2021, as a percentage of the total area.

Figure 5 shows a sample of settlement expansion in eMbalenhle township from 1986 to 2021. There are many settlements within the catchment, with differing rates of expansion during the study period.

Regarding the remaining classes (mining, water body and woody vegetation), there were different trends in area changes; for example, there was a noticeable increase in the mining area, except for the periods 1993–2000 and 2007–2014. The change detection did not show any trends for the water body class, but it showed that the woody vegetation areas decreased from 1986 to 2000 and expanded from 2000 to 2014 before it decreased again in 2021 (see Figure 6 and Table 7). The woody vegetation class was particularly noticeable in the mountainous parts on the east and southeastern boundaries of the catchment.

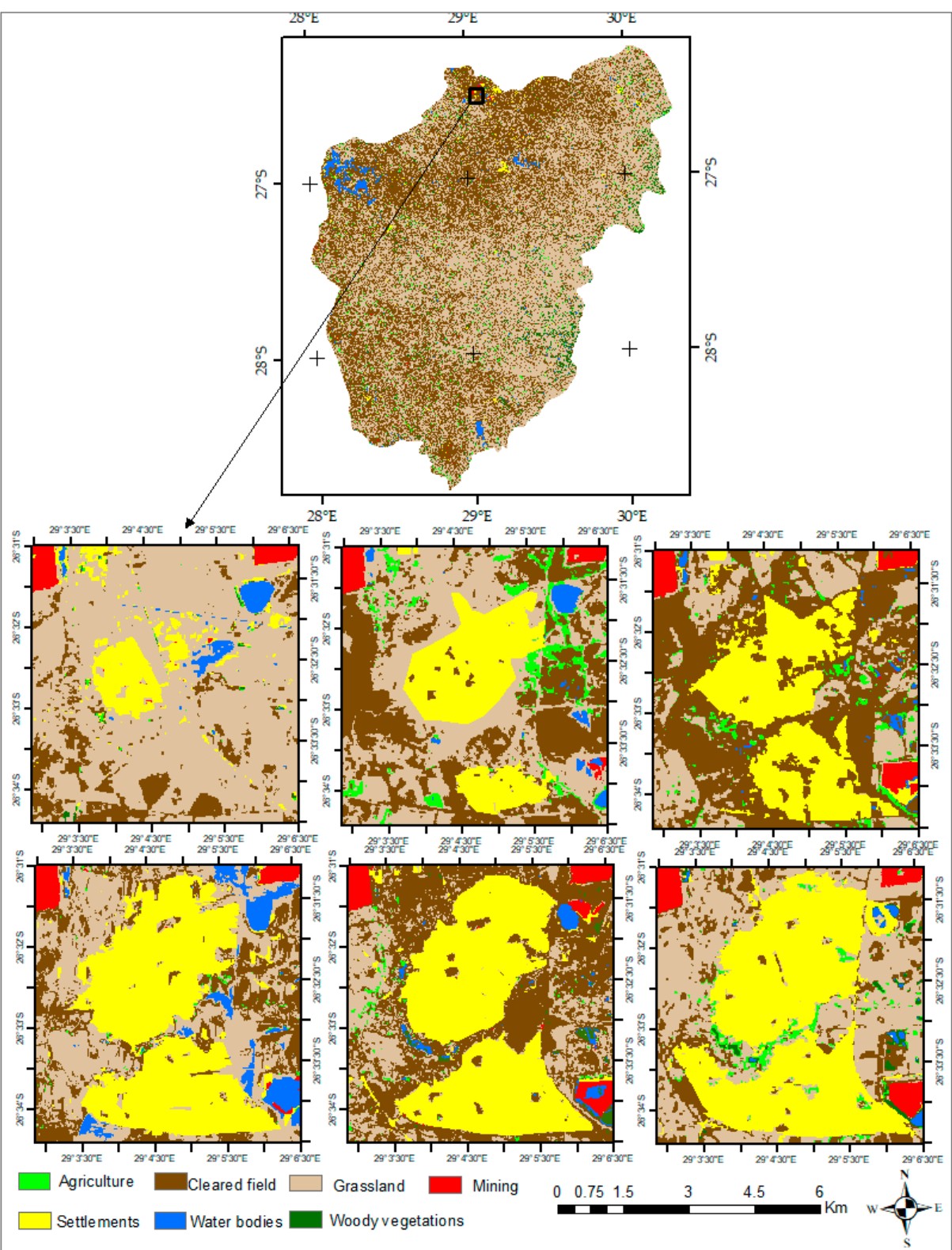

**Figure 5.** A sample of settlement expansion for eMbalenhle township in the northern part of the Vaal Dam Catchment during the period 1986–2021.

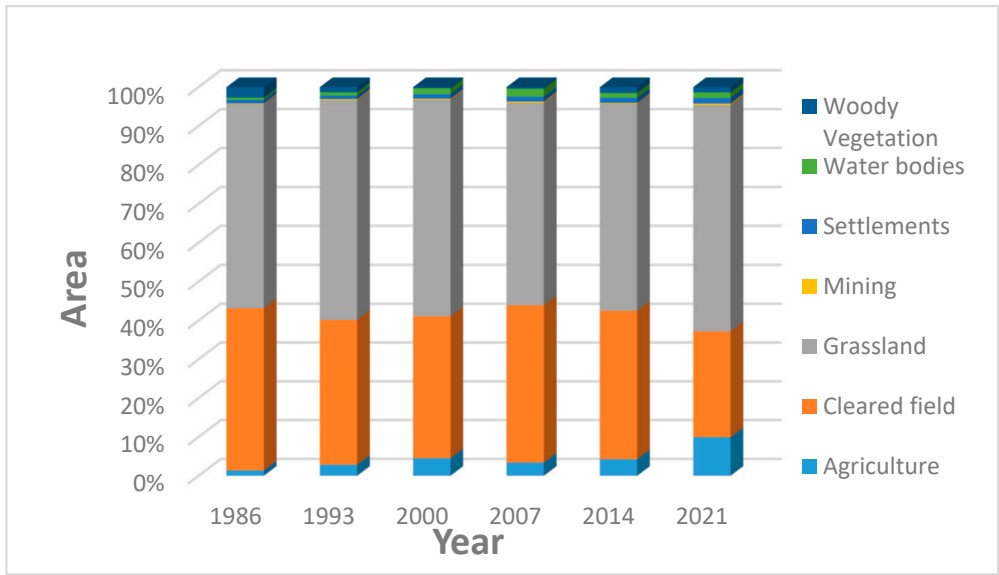

**Figure 6.** Land use land cover (LULC) change for 1986, 1993, 2000, 2007, 2014 and 2021 in the Vaal Dam Catchment.

## 4. Discussion

The LULC classification results showed the domination of the grassland class (including the land used for cattle and sheep grazing farms and pastures) in the study area, followed by the land used for agricultural activities (that is, the summation of agriculture and cleared field classes). The changing pattern of these two categories showed different rates of increase and decrease, but, contrary to what was expected, no significant ongoing increase was noticed for land used for agricultural activities. Although the settlement area class consists of only a relatively small proportion of the total study area, it showed continuous expansion from 1986 to 2021. In contrast, the mining land cover class showed varying patterns of increase and decrease, and varying patterns were also noticed for the remaining classes.

The study successfully achieved its aim, in that it mapped the changes and patterns in LULC and associated dynamics for this strategically important area of South Africa over the 35 years ending in 2021. The most important land cover change within the catchment was the expansion of settlement areas related to the economic activities within the area in recent decades. Many gold and coal mines [16], as well as synthetic fuel complexes, were constructed and operated in the late 1970s [18]. This economic growth significantly increased the expansion of towns associated with these developments (see Figure 4 and Table 7). Figure 5 focused on eMbalenhle township and is a sample of this settlement expansion. The latter township was established in 1978 to increase accommodation needs for the people working for the synthetic fuel manufacturing plant at Sasol. The expansion of many settlement areas within the catchment owing to population growth and their need to grow more food, the construction of houses and industries, etc., have contributed both directly and indirectly to environmental, ecosystem and water quality degradation. In most of the previous studies conducted, water quality and ecosystem degradation have been identified as the most concerning problems within the catchment [16,18,27,30].

The relative area used for agricultural activities showed no significant changes, even though it was expected to expand during the study period to meet population growth and their increasing needs for food. In fact, according to Biggs (2002), privately owned farms occupy around 68% of South Africa's land area [44]. This indicates that the extent of privately owned farmland remains relatively stable [44], and only limited areas are available for agricultural expansion [32]. Nevertheless, increasing productivity using modern techniques and soil fertilisers may have been applied to increase the productivity per unit hectare [44].

The results of LULC classification using RF classifier [45] showed good accuracy in mapping the Vaal Dam Catchment. It has provided much more useful details for the study area than those already published on global land cover patterns (https://lcviewer.vito.be/download, http://maps.elie.ucl.ac.be/CCI/viewer/download.php and http://www.globallandcover.com/home) accessed on 8 January 2023. The settlement class was clearly detected with all classification schemes; they showed high similarity between them when comparing same-year maps. The remaining classes appeared more generalised in the published global schemes while they were well-detected in this study. Results of this comparison confirm that the method used in this study gives reliable, highly accurate LULC classification results and can be adopted in other regions. However, this methodology has some limitations in some land cover categories, such as the mining class. The detected mining class in this study represents rock and ash dumps and piles from gold and coal mining; only highly reflective materials of those dumps and piles were detected, while many of the mining sites contained small dams and materials for which the colours could not be detected. They may thus have been misclassified. Furthermore, acid mine drainage within or around the study area from abandoned and active mine sites potentially is causing severe water quality and environmental issues but this could not necessarily be detected in this study [20,22]. With the Landsat data (30-m spatial resolution) methodology applied, it is not easy to detect the narrow surface drainage ditches carrying acid mine effluences. The same limitation can be considered regarding mapping the uncontrolled seepage and flooding of sewage in some settlement areas within the catchment. This situation poses risks to public health, ecosystems and water quality and has received extensive coverage by media channels in recent years (https://www.groundup.org.za/article/sewage-seeps-into-vaal-dam-as-mpumalanga-water-treatment-plants-fail/) accessed 27 September 2022. Another concerning issue is that the water-body areas detected included wastewater storage dams (such as Leewpan near to eMbalenhle) [18]. Consideration should be given to the possibility of such dams contaminating nearby water resources.

The chosen study area is characterised by relative homogeneity of its dominant land cover classes; this made it possible to map the area with RF classifier using only a small but representative training data sample (see Section 2.3.3). As shown by Ebrahimy et al. (2021), the RF approach is successful in land cover mapping with limited reference sample data [46].

## 5. Conclusions

This study used Landsat data series to assess LULC changes in the Vaal Dam Catchment area over 7-year intervals for the period 1986–2021. A random forest classifier method consisting of 500 trees was used owing to its advantages over most of the other classifiers in LULC detecting. It was used to classify six mosaicked images covering the study area in seven land cover classes: namely, agriculture, cleared fields, grassland, mining, settlements, water bodies and woody vegetation. The results of the classification reveal the following percentage composition for the total study area for the period investigated: grasslands covered between 52.0% and 56.9%, cleared fields ranged between 28.6% to 41.8%, agriculture covered 1.4% to 10.0%, mining ranged between 0.14% and 0.39%, settlements covered between 0.87% and 1.43%, water bodies ranged between 0.65% and 2.10% and woody vegetation ranged between 0.29% and 2.66%. The results showed varying patterns of change in LULC (both increases and decreases) observed in most of the classes except for the settlements class. The latter showed a clear increasing trend from 1986 to 2021 resulting from economic development within the area in the last few decades. The OA of the classification of the various LULC types ranged between 91% and 97%. The RF models reported an average OOB error that ranges from 3.75% to 8.98%. The area was dominated by the grassland class for the study period, followed by cultivation land use (which includes the agriculture and cleared field classes). The generated maps provide spatial and temporal patterns of land cover and the changes for the periods studied. An ongoing rapid increase in population growth will have even more significant effects on

the region's environment and economic spheres. Therefore, it is of prime importance that these developments be carefully considered in this important catchment as it is one of the primary sources of water for the Vaal Dam, which supplies more than 13 million people in the Gauteng and Mpumalanga provinces. Such studies can support efforts to protect ecosystem functioning and water resources from further deterioration in water quality.

**Author Contributions:** Conceptualisation, A.O.; methodology, A.O. and E.A.; software, A.O., E.A. and K.A.A.; validation, A.O., E.A. and K.A.A.; formal analysis, A.O.; investigation, A.O.; resources, A.O.; data curation, A.O.; writing—original draft preparation, A.O.; writing—review and editing, A.O., E.A. and K.A.A.; visualization, A.O., E.A. and K.A.A.; supervision, E.A. and K.A.A. All authors have read and agreed to the published version of the manuscript.

**Funding:** This research received no external funding.

**Institutional Review Board Statement:** Not applicable.

**Informed Consent Statement:** Not applicable.

**Data Availability Statement:** The data will be available upon request.

**Acknowledgments:** The authors would like to acknowledge the support of Lyn Brown for her valuable language proofreading of the manuscript, and we would like to show their appreciation to the reviewers and the editor for their valuable comments and suggestions.

**Conflicts of Interest:** The authors declare no conflict of interest.

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
