# Peer review of "Land Use and Land Cover Change in the Vaal Dam Catchment, South Africa: A Study Based on Remote Sensing and Time Series Analysis"

_2673-7418, doi:10.3390/geomatics3010011_

Round 1

Reviewer 1 Report

The authors performed the classification of a LUCC and further analyzed the time series of the classification results. There are some comments that exist.

1. The introduction lacks a discussion of existing research and does not address the gap that this study can address. the authors need a fuller literature review to formulate the research questions.

2. Machine learning models require a large number of samples, however, there are only about 100 sample points for each land use type used for modeling in the paper, which may cause overfitting problems.

3. The authors should give a detailed explanation of how the hyperparameters of the random forest model are chosen. On which dataset was it acquired?

4. It is recommended to add a technical roadmap to clarify the framework of this paper.

5. The minutes and seconds whose values are all 0 can be removed from the latitude and longitude of all maps.

Author Response

Dear Reviewer 1

I would like to thank you for your valuable comments and suggestions, please find in the attached table my responses to your comments.

Kind regards

Reviewer 2 Report

Dear authors!

We were pleased to review the manuscript “Land Use and Land Cover Change in the Vaal Dam Catchment, South Africa: A Study Based on Remote Sensing and Time Series Analysis”. The article is devoted to the topical issue of LULC change on the regional scale of the study. The authors obtained the final results, but it is partially unclear how exactly they were obtained. The research methodology requires clarification and detailing. In general, the article has a large number of inaccuracies.

Answers to the comments below will help to significantly improve the article:

1. What does “+/- 3%” mean on line 25? Is this the value of the calculation error? If yes, please provide such values for the values presented in line 25.

2. Lines 65-66 indicate "Previous studies have used satellites such as the..." but do not provide references to the results of previous studies. Similar remarks are found throughout the text. Check the need for references to the literature in the text of the work.

3. Lines 43-84 would be more appropriate in section 2 "Materials and Methods". Lines 43-60 contain a description of the study area, and lines 60-84 contain a description of the space images used and research methods.

4. At the same time, if lines 60-84 will be moved to section 2 "Materials and Methods", then section 1 "Introduction" must be significantly expanded. To do this, conduct a broader review of the literature. Give examples of the use of LULC in other regions of the world. Show that the use of LULC is actively used in Africa and South Africa. And if not, then indicate that your research is the latest. Indicate why it is important to use remote sensing methods, what advantages it brings for the economy and solving environmental problems.

5. In lines 105-106 there is no caption to the title of Figure 1.

6. In figure 1 on the map in the upper left corner - please show the countries bordering South Africa, sign the seas and oceans, indicate the scale, degree grid, north direction. On the map, on the right - please sign the names of the rivers along the riverbeds.

7. Please specify the method by which the selection of Vaal Dam catchment was made.

8. Please specify in which program the calculations were made? What method was used for radiometric and atmospheric correction?

9. Line 113 probably misses 2007?

10. Please provide references where did The South African National Land-Cover data (line 126) come from?

11. It is not entirely clear from the text of the article what is presented in Table 3. Please describe the methodology in more detail. Why did you use RF classifier? What are its advantages over Image classification using the ArcGIS Spatial Analyst? Or other software solutions?

12. In general, the research methodology is written extremely fragmented. Indicate in which programs you worked. It is not clear how 2.2.1, 2.2.2., 2.2.3, 2.2.4 and 2.2.5 agree with each other. Please describe the method in more detail. Step by step. If you have worked in GIS, then perhaps provide a graphical scheme of the study or a study model (for example, Model Builder ArcGIS, etc.). Without this, it will be extremely difficult for the reader to reproduce your methodology for studying any other region of the world.

13. Specify why your total area of the study area changes in Table 5 (Line - Total) from 1986 to 2021.

14. On line 271 in the figure, please add the name of the vertical axis on the graph.

15. In Figure 4, please indicate the scale for each map. You probably have an error in the figure - since one scale is given for 7 maps. Bring the degree grid.

16. The “Discussion” section should be reworked and separate information containing specific research results from it (place it in the “Results” section) (for example, lines 358-361). Give a comparison of your data with the work of other authors. Compare your results with other areas in South Africa/Africa/world. In general, giving good results of the study, the authors do not answer the question - what is the root cause of these changes? Perhaps it is necessary to take into account the impact of climate change, or are these changes caused only by economic activity, or perhaps a number of other reasons? Or maybe the cumulative action of any causes together? This will help the reader to better understand what you have done.

17. Maybe your Land cover data is better than already published https://lcviewer.vito.be/download, http://maps.elie.ucl.ac.be/CCI/viewer/download.php, http://www.globallandcover.com/home_en.html? Please provide a comparison in the "Discussion" section.

The article can be accepted after correction of remarks.

Author Response

Dear Reviewer 2

Thank you for you valuable comments and suggestions, they are really improved the manuscript. Please find the attached file my responses to you comments.

Kind regards

Reviewer 3 Report

In this manuscript, a change detection approach based on post-classification comparison is presented. I think it is necessary to improve the quality of the manuscript according to suggestion:

1.       Literature review: it is necessary to mention some related papers in the introduction. Please discuss them from viewpoints of advantages and limitations.

2.       Aims: I cannot understand novelty of this study. It must be mentioned clearly in the end of the introduction section.

3.       Please provide a workflow in the method section.

4.       Line 92: please define units.

5.       It is necessary to mention reasons for selecting steps in the methodology. For example, why random forest?

6.       Which bands of satellite images were used?

7.       Table 3: I am not sure that number of samples is enough for a robust classification.

8.       Table 4: what is the best performance? What is its unit?

9.       Please apply two other classifiers like maximum likelihood to compare results with RF.

10.   There are some typo and grammatical errors in the text that should be revised carefully.

Author Response

Dear Reviewer 3

Thank you for your valuable comments and suggestions, please find in the attached table my response to your comments.

Kind regards

Round 2

Reviewer 1 Report

My concerns and suggestions were well-addressed.

Author Response

Dear Reviewer

Thank you so much for your valuable comments.

kind regards

Reviewer 2 Report

Dear authors.

I am grateful to you for the quick answers to the questions.

 The quality of your manuscript has improved significantly.

There are a few minor comments left.

1. In lines 28-29, add keywords.

2. In Figure 4, specify the dimension of the vertical axis (perhaps square km?)? Sign the name of the horizontal axis.

The article can be accepted after minor revision

Author Response

Dear Editor

I would like to thank you again for your valuable comments, The keywords have been added to specified lines. The vertical axis dimension and the name of the horizontal axis have been added to figure 4.

Reviewer 3 Report

1- My Q:  In Table3, I am not sure that number of samples is enough for a robust classification.

( Please provide some references to prove your claim)

*****************

2-  Aims: I cannot understand novelty of this study. It must be mentioned clearly in the end of the introduction section.

(I still believe that only changing the location of the study area can not be consider as innovation, unless you also use different methods.)

RF classifier works in small sample numbers as well as large samples without overfitting

**************

This is the first study covers hole Vaal Dam Catchment in term of LULC mapping using remote sensing data..

Author Response

Dear Editor

Thank you again for your valuable comments, please see the attached file; containing our response to your comments.

best regards
